posttraumatic stress disorder; genomics; global health; trauma; neurobiology

**Corresponding author:**
Jacqueline Womersley;
Email: jsw1@sun.ac.za

# Advances in the molecular neurobiology of posttraumatic stress disorder from global contexts: A systematic review of longitudinal studies

Jacqueline S. Womersley[1,2] , Morne du Plessis[1,2], M. Claire Greene[3] ,
Leigh L. van den Heuwel[1,2], Eugene Kinyanda[4,5] and Soraya Seedat[1,2]

[1]Department of Psychiatry, Stellenbosch University, Cape Town, South Africa; [2]South African Medical Research Council/Stellenbosch University Genomics of Brain Disorders Extramural Unit, Stellenbosch University, Cape Town, South Africa; [3]Program on Forced Migration and Health, Heilbrunn Department of Population and Family Health, Columbia University Mailman School of Public Health, New York, NY, USA; [4]MRC/UVRI and LSHTM Uganda Research Unit, Entebbe, Uganda and [5]Department of Psychiatry, College of Health Sciences, Makerere University, Kampala, Uganda

## Abstract

Trauma exposure is prevalent globally and is a defining event for the development of posttraumatic stress disorder (PTSD), characterised by intrusive thoughts, avoidance behaviours, hypervigilance and negative alterations in cognition and mood. Exposure to trauma elicits a range of physiological responses which can interact with environmental factors to confer relative risk or resilience for PTSD. This systematic review summarises the findings of longitudinal studies examining biological correlates predictive of PTSD symptomology. Databases (Pubmed, Scopus and Web of Science) were systematically searched using relevant keywords for studies published between 1 January 2021 and 31 December 2022. English language studies were included if they were original research manuscripts or meta-analyses of cohort investigations that assessed longitudinal relationships between one or more molecular-level measures and either PTSD status or symptoms. Eighteen of the 1,042 records identified were included. Studies primarily included military veterans/personnel, individuals admitted to hospitals after acute traumatic injury, and women exposed to interpersonal violence or rape. Genomic, inflammation and endocrine measures were the most commonly assessed molecular markers and highlighted processes related to inflammation, stress responding, and learning and memory. Quality assessments were done using the Systematic Appraisal of Quality in Observational Research, and the majority of studies were rated as being of high quality, with the remainder of moderate quality. Studies were predominantly conducted in upper-income countries. Those performed in low- and middle-income countries were not broadly representative in terms of demographic, trauma type and geographic profiles, with three out of the four studies conducted assessing only female participants, rape exposure and South Africa, respectively. They also did not generate multimodal data or use machine learning or multilevel modelling, potentially reflecting greater resource limitations in LMICs. Research examining molecular contributions to PTSD does not adequately reflect the global burden of the disorder.

## Impact statement

Though most people experience at least one traumatic event in their lifetime, only a subset go on to develop PTSD. Biological mechanisms play an important role in determining risk and resilience. Advances in molecular technologies and data analytic procedures now provide unprecedented insights into the molecular aetiology underlying PTSD. Ideally, identification of biological correlates of PTSD should be used to stratify trauma-exposed individuals according to risk, target preventative measures and interventions accordingly, identify biological targets for therapeutic modulation and track treatment response. This systematic review provides a synthesis of the evidence base globally, highlights key mechanisms and approaches used in molecular research, and compares and contrasts the nature and form of the literature base in upper- and low- and middle-income countries. Studies drawing on different biological markers, including genotypic, epigenetic, transcriptomic, endocrinological and serum level data, consistently point to the role of the stress response, inflammation, and learning and memory in PTSD symptomology. Contingent risk granted by these mechanisms depends on environmental and demographic factors. Our search results indicate a mismatch between the global trauma burden and the research conducted, with studies primarily conducted in upper-income countries. Detailed investigations of the molecular mechanisms underlying PTSD in diverse populations and contexts are required if the promise offered by biological insights is to be globally relevant, actionable and equitable.

## Introduction

Trauma exposure is prevalent worldwide with approximately 70% of people reporting exposure to at least one traumatic event in their lifetime (Benjet et al., 2016). Based on analysis of 26 population surveys, around 5.6% of trauma-exposed individuals will go on to develop posttraumatic stress disorder (PTSD), which is characterised by symptoms of hyperarousal, reexperiencing, avoidance and negative alterations to cognition and mood (American Psychiatric Association, 2013; Koenen et al., 2017). Though PTSD contributes to poor mental health globally, low- or middle-income countries (LMICs) are disproportionately affected. Not only is the total population of LMICs substantially higher than that in upper-income countries (UICs), but these regions also suffer from a dual burden of potent stressors and limited mental health care resources (Purgato and Olff, 2015). For example, the age-standardised prevalence of PTSD in conflict settings is approximately 15.3% and this burden is primarily in LMICs (Charlson et al., 2019). Hoppen et al. estimated that in 2019, more than 99% of adults who had experienced war in the preceding 30 years resided in LMICs, which by extrapolation accounts for approximately 3.1 million PTSD-associated disability-adjusted life years in these countries (Hoppen et al., 2021). In addition, factors, such as higher socioeconomic status, living standards, community infrastructure and use of mental health services, which protect against the development of posttraumatic stress (PTS) following disasters and pandemics may be less prevalent in LMICs (Newnham et al., 2022).

### Neurobiology of traumatic stress outcomes

Biological mechanisms grant contingent risk or resilience and contribute to the considerable interindividual variability in posttraumatic stress (PTS) symptom presentation, severity, trajectory and treatment response (Ressler et al., 2022). Translational neuroscience provides evidence for genomics, neural circuitry and neurotransmission aberrations and dysregulated immune system processes as causes or correlates of PTSD. PTSD is unique among disorders in requiring a precipitating traumatic exposure, which provides an opportunity for early preventative or ameliorative interventions. Identification of biological correlates of PTSD symptom trajectories has the potential to elucidate underlying biological mechanisms, stratify individuals according to relative risk, target and track the efficacy of treatment, and enable more accurate assessment of novel intervention strategies (Schultebraucks et al., 2021).

### Aims of the systematic review

The aim of this systematic review is to summarise recent insights into the molecular aetiology and pathophysiology of PTSD, identify key biological signatures involved, highlight methodological advances, and provide an overview of the global nature, state and representation of the research field. We focussed on longitudinal studies, as they can more accurately capture the complex and dynamic relationships between biological signatures and symptom trajectories and can assess the role of potentially modifiable factors as moderators or mediators of risk. To provide a more in-depth review, we additionally chose to focus on studies examining biological correlates as predictors of PTSD risk or symptom trajectory, rather than treatment response.

## Methods

The Preferred Reporting Items for Systematic Reviews and Meta-Analyses guidelines were employed in conducting the systematic review (Page et al., 2021).

### Search strategy

Two researchers, JSW and MdP, independently searched the Pubmed, Scopus and Web of Science databases on 9 September 2022 for manuscripts published since 1 January 2021 using the following search terms: *(("post-traumatic stress" (All Fields)) OR ("posttraumatic stress" (All Fields)) OR (PTSD (All Fields))) AND ((neurobiolog\* (All Fields)) OR (genom\* (All Fields)) OR (DNA (All Fields)) OR ("stress hormone" (All Fields))) AND (English(Language)) NOT (animal)*. These terms were designed to capture studies examining molecular correlates (including genomic, endocrine and inflammatory measures) of PTSD in human participants. The term 'trauma' was not included due to its high overlap with studies in the fields of surgery, physical rehabilitation and orthopaedics, which yield datapoints outside the scope of the review. The search was repeated using the same strategy on 22 March 2023 to include articles published up until 31 December 2022. This two-year time frame was chosen to focus on recent findings so that the review provides state-of-the-art insights into PTSD.

### Eligibility criteria and selection process

The online open-source software Rayyan was used to collate the database search outputs, scan for duplicates and conduct a preliminary selection of articles (Ouzzani et al., 2016). Suspected duplicate articles flagged by the software were manually checked and removed if necessary. Selection of publications was performed by applying inclusion and exclusion criteria to the study abstracts. Studies to be included had to (a) comprise original research manuscripts or meta-analyses of (b) cohort investigations that (c) assessed longitudinal relationships between (d) one or more molecular-level measures and (e) either PTS symptoms or PTSD status, and (f) be written in English. Exclusion criteria related to the manuscript type and research approach employed. Narrative reviews, case studies and protocol papers, as well as studies of exclusively healthy participants exposed to an experimental stressor or using animal models were excluded. The final study selection was based on a review of the full text manuscripts. A third researcher was approached when inclusion decisions were discrepant, and a final decision was reached by consensus. A total of 18 manuscripts were selected for the systematic review (Figure 1).

### Quality assessment

The quality of each study was independently assessed by researchers (JSW, MCG, MdP) using the Systematic Appraisal of Quality in Observational Research (Ross et al., 2011). This tool evaluates the quality of evidence-based cohort and case–control observational studies in psychiatric research and assesses quality across five categories, namely sample, control/comparison group, measurement and output quality, follow-up, and distorting influences. The extent to which each study complies with the two to five statements listed under each category is used to classify quality as adequate, unclear or inadequate. A final quality level ranging from low to high was assigned based on these metrics.

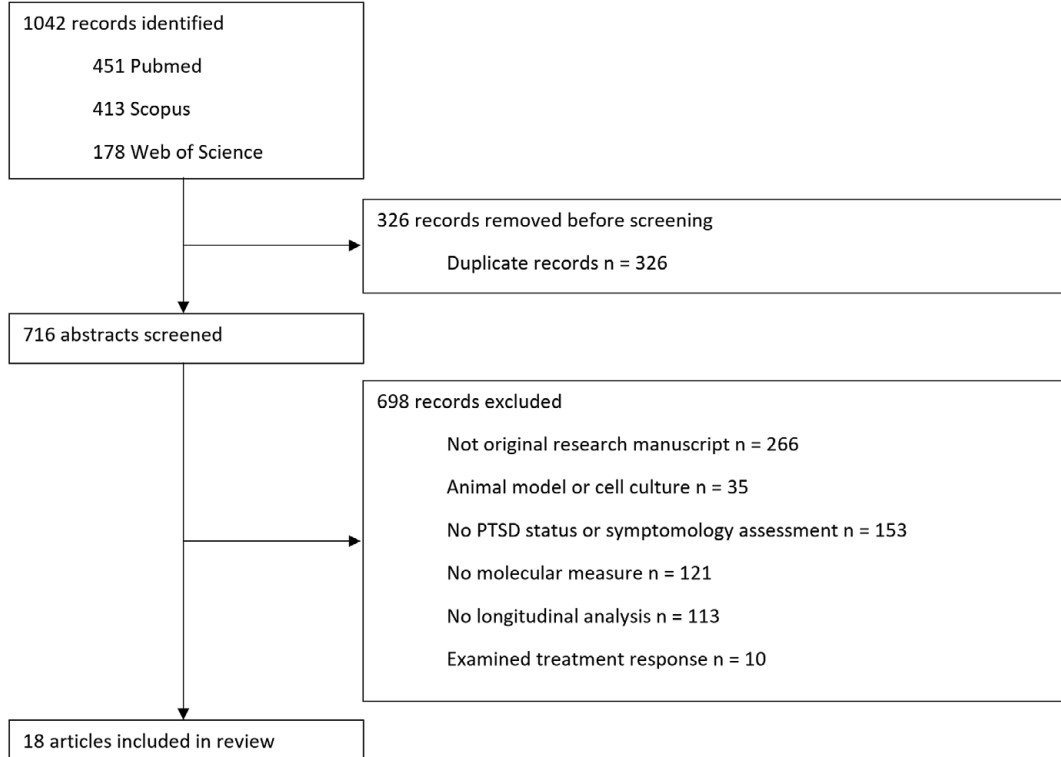

**Figure 1.** Study selection flow diagram. Molecular biology and PTSD.

### Data extraction

Researchers (JSW, MdP, MCG) individually extracted data from articles using a template designed to capture information relevant to the review aims. The study design and setting category reports on the design, location of the study and/or site of participant recruitment, and the name of the parent study or cohort if applicable. The sample characteristics category records the number, sex, age and ancestry of participants broken down according to experimental group or study cohorts if necessary. The third category for data extraction was the overall stated aim of the study. Clinical measures record the PTSD assessment instruments, while the type of molecular measure and source tissue were included under molecular measures. The sixth category, primary findings, captured the main outcomes of the stated study aims. Findings unrelated to the relationship between molecular measures and PTSD were excluded. Extracted data are reported in Table 1.

### Results

#### Study characteristics

Our search yielded 18 studies that examined longitudinal relationships between PTS/PTSD and molecular biology measures with sample sizes ranging from 39 to 1,135 participants. Study groups were primarily comprised of military veterans/personnel (n = 4), individuals hospitalised following traumatic injury (n = 4), or women exposed to interpersonal violence/trauma (n = 3) or rape (n = 3). The Clinician-Administered PTSD Scale ($n_{DSM-IV}$ = 5, $n_{DSM-5}$ = 2), PTSD Checklist ($n_{Civilian\ version}$ = 3, $n_{DSM-5}$ = 4, $n_{DSM-IV}$ = 2) and Davidson Trauma Scale (n = 3) were the most

commonly used measures with both continuous scores and case–control status used as outcome variables. Genomic investigations were the most frequently employed molecular measure and included gene expression (n = 2), genotype (n = 9) and DNA methylation (n = 7) analyses that spanned candidate and whole-genome approaches. Endocrine (n = 2) and inflammatory marker (n = 2) measures were also represented. Four of the studies were performed in LMICs, of which three were conducted in South Africa. The majority of upper-income country (UIC) studies were conducted in the USA (n = 10). Of the 15 studies that reported on ancestry/ethnicity, White/European/Caucasian and African American/Black ancestry were the sole or most prevalent ancestral grouping in seven studies each.

#### Molecular signatures of PTSD/PTS

##### Candidate genotype or marker investigations

Several studies employed candidate gene approaches. A study of mother–child dyads found that, in children carrying the 'risk' T allele of rs1360780, a variant found in the stress-related FKBP5 gene, higher maternal PTS scores at baseline predicted more severe total and dissociation-related trauma symptoms in children 9 months later. This relationship remained significant even when trauma exposure in the children was controlled for (Pereira et al., 2021). Landoni et al. (2022) examined the influence of the serotonin transporter long polymorphic region on the development of perinatal PTSD assessed in the 2 months prior to birth (T1), as well as at 2–3 weeks (T2) and 3–4 months (T3) postnatally. The lower expression SS polymorphism was associated with elevated T2 intrusive and T3 hyperarousal symptoms, and moderated the worsening of intrusive, hyperarousal and avoidance symptoms

**Table 1.** Data extracted from studies examining the molecular contribution to PTSD

| Study | Design and setting | Sample characteristics | | | | Aims | Clinical measures | Molecular measures | Primary findings |
|---|---|---|---|---|---|---|---|---|---|
| | | Sample size | Age | Sex (female) | Ancestry/ ethnicity | | | | |
| Chen et al. (2022) | Longitudinal study of Chinese high school students following the Wenchuan earthquake | n = 462 | Grades 11 & 12 | 57% | Not specified | To examine the longitudinal associations between PTSD, environmental factors, and IL-10 rs1800872 genotype | PCL-C at 6-, 12- and 18 months post exposure | IL-10 rs1800872 determined using DNA extracted from whole blood 6 months post exposure | Male AA homozygous participants had higher PTSD prevalence relative to male C allele carriers at 18 months. The prevalence of PTSD in the sample reduced from 6 to 18 months, except among males carrying the AA genotype. Patterns of PCL-C scores were associated with genotype |
| Cordero et al. (2022) | Swiss prospective longitudinal study of mothers exposed to interpersonal violence and their children recruited to the Geneva Early Childhood Stress Study | DNA methylation group $n_{Total}$ = 48 mother–child dyads $n_{PTSD}$ = 26 mother–child dyads DNA methylation and child behaviour group n = 36 mother–child dyads | Children at baseline: 27.8 ± 8.8 months Children at follow-up: 7.5 ± 0.95 years | Mothers: 100% Children: 69.4% | Not specified but recruitment in Switzerland with French language proficiency required | To examine NR3C1 methylation in mothers with interpersonal violence-associated PTSD and their children, and to investigate the relationship between maternal methylation profile and child psychopathology | CAPS-IV (mother) | Bisulphite pyrosequencing of DNA extracted from saliva collected from mothers and their 12–42 month-old children. Control for sample cell composition not reported. | Infant and maternal *NR3C1* methylation is significantly correlated only in mothers with a diagnosis of PTSD. Lower maternal methylation predicted later increased externalising behaviours in children at 5-9 years of age. |
| deRoon-Cassini et al. (2022) | Prospective longitudinal study of adults hopsitalised following traumatic injury in the USA | n = 170 | 42.8 ± 16.5 years | 31% | Caucasian 47.1%, Black American 44.1%, Hispanic/Latinx 7.6%, American Indian/Alaskan Native 1.2% | To examine associations between endocannabinoid levels, genetic polymorphisms, and PTSD in adults hospitalised for traumatic injury | CAPS-5 and PCL-5 at baseline and 6-8 months follow-up | *FAAH* rs324420 genotype measured in DNA extracted from whole blood. Circulating endocannabinoid and cortisol levels measured in serum at baseline and 6-8 months follow-up | Higher levels of AEA were associated with higher PTSD symptom levels at follow-up. Higher levels of 2-AG were associated with higher PTSD symptom levels at follow-up among minorities only. The rs324420 AA genotype was associated with higher PTSD symptom levels in Black participants only. |

(*Continued*)

| Study | Design and setting | Sample characteristics | | | | Aims | Clinical measures | Molecular measures | Primary findings |
|---|---|---|---|---|---|---|---|---|---|
| | | Sample size | Age | Sex (female) | Ancestry/ ethnicity | | | | |
| Hawn et al. (2022) | Cross-sectional and longitudinal study of US military veterans recruited to the TRACTS study | Cross-sectional n = 478 Longitudinal n = 296 | Cross-sectional 32.92 ± 8.81 years Longitudinal 33.25 ± 9.20 years | Cross-sectional 9.6% Longitudinal 9.8% | Cross-sectional White 74%, Black/African American 9%, Asian 3%, American Indian/Alaska Native <1%, Hawaiian/other Pacific Islander <1%, Unknown 15%, Longitudinal White 74%, Black/African American 10%, Asian 2%, American Indian/Alaska Native <1%, Hawaiian/other Pacific Islander <1%, Unknown 16% | To determine whether *AIM2* methylation mediates the cross-sectional and longitudinal relationships between PTSD symptom severity and peripheral markers of inflammation and neuropathology at baseline and 2-year follow-up | CAPS-IV / CAPS-5 at baseline and 2-year follow-up. | *AIM2* methylation assessed using DNA extracted from blood. Cellular heterogeneity was accounted for in analyses. Measurement of 7 peripheral markers of neuropathology (NfL, GFAP, tau, Aβ40, Aβ42, pNfH and NSE). 5 inflammatory markers: IL-6, IL-10, eotaxin, TNFα and CRP. Measurements performed at baseline and approximately 2 years follow-up. | *AIM2* methylation mediated the participant between PTSD symptom severity and peripheral markers of participant (IL-10, IL-6, and TNFα) and neuropathology (NfL and GFAP) at baseline. PTSD symptom severity positively associated with levels of IL-6 and CRP and inversely associated with cg10636246 methylation at baseline. *AIM2* methylation mediated the association between baseline PTSD symptom severity and follow-up IL-10. |
| Katrinli et al. (2022) | Longitudinal meta-analysis study of participant sourced from three military cohorts: MRS (USA), Army STARRS (USA) and PRISMO (Netherlands) | $n_{Total}$ = 429 $n_{PTSD}$ = 199 $n_{Controls}$ = 230 | MRS: 22.06 ± 2.24 STARRS: 24.47 ± 4.84 PRISMO: 27.35 ± 9.06 | 0% | Total: 77% European ancestry MRS: European 69.3%, African American 3.9%, Latino 12.6%, East Asian 1.6%, other 12.6% STARRS: European 67.4%, African American 11.4%, other 21.2% PRISMO: European 100% | To conduct a meta-analysis assessing epigenome-wide methylation changes associated with PTSD symptom severity pre and post-deployment, and to identify DNA methylation profiles associated with baseline and longitudinal PTS symptom severity in military personnel | MRS: 17-item PCL-IV pre-and post-deployment Army STARRS: 6-item PCL pre and 17-item PCL-IV post-deployment PRISMO: SRIP pre and post-deployment | Genome-wide DNA methylation assessed in DNA extracted from whole blood. Cellular heterogeneity was accounted for in analyses. | 4 DMPs and 88 DMRs were associated with PTSD symptom severity. Change in 15 DMRs was associated with change in PTSD symptom severity. |
| Landoni et al. (2022) | Longitudinal study of perinatal depressive and PTSD symptoms in Italian women | $n_{enrolled}$ = 155 $n_{prenatal}$ = 141 $n_{postnatal1}$ = 127 $n_{postnatal2}$ = 110 | 33.37 ± 4.64 years | 100% | 100% Caucasian | To examine the association between the *5-HTTLPR* polymorphism and the development of perinatal depressive and PTSD symptoms | LASC prenatally, and LASC and PPQ at 2-3 weeks and 3 months postnatal | *5HTTP-LPR* polymorphism assessed in DNA extracted from whole blood or saliva | The SS genotype was associated with worse PTSD symptom severity at 2-3 weeks and 2-3 months postpartum. |

**Table 1.** (Continued)

| Study | Design and setting | Sample characteristics | | | | Aims | Clinical measures | Molecular measures | Primary findings |
|---|---|---|---|---|---|---|---|---|---|
| | | Sample size | Age | Sex (female) | Ancestry/ ethnicity | | | | |
| Lori et al. (2021) | Prospective longitudinal cohort study of participants admitted to the Atlanta Grady Memorial hospital Emergency Department in the USA | PTSD trajectory analysis $n_{Total}$ = 224 $n_{Resilient}$ = 130 $n_{Chronic}$ = 23 $n_{Remitting}$ = 71 Trascriptomic analysis $n_{Total}$ = 153 $n_{Chronic}$ = 23 $n_{Resilient}$ = 130 | PTSD trajectory analysis Total: 35.8 ± 12.5 years Resilient: 35.9 + 12.4 Chronic: 34.6 + 11.2 Remitting: 36.0 ± 13.2 Transcriptomic analysis Chronic: 34.6 + 11.2 Resilient: 35.9 + 12.4 | Total: 50.9% Resilient: 47.7% Chronic: 60.9% Remitting: 53.5% | Total: African 71.4%, European 19.6%, other 8.9% Resilient: African 67.7%, European 22.3%, other 10% Chronic: African 73.9%, European 13%, other 13% Remitting: African 77.5%, European 16.9%, other 5.6% | To determine whether blood-based transcriptomic biomarkers collected shortly after an index trauma could predict PTSD symptom trajectories over subsequent months | PSS-I at 1, 3, 6 and 12 months | Transcriptome, eQTLs and meQTLs assessed in DNA extracted from whole blood at baseline. Cellular heterogeneity was accounted for in analyses. | Expression of *GRIN3B* and *AMOTL1*, differed between chronic and resilient PTSD trajectories. The finding of low *GRIN3B* expression as a predictor of resilience survived Bonferroni multiple testing correction and subsequent adjustment for drug use, alcohol and education. |
| Mehta et al. (2022) | Cross-sectional and longitudinal study of paramedicine students in Australia | n = 39 | 23.44 ± 1.08 years (mean ± standard error) | 61.50% | Caucasian 89.7% | To investigate biological ageing using epigenetic clocks in relation to trauma exposure, and to examine environmental factors influencing this relationship | PCL-5 at baseline and 1 year | DNAm GrimAge, Horvath and BloodSkinAge epigenetic clocks measured using whole-genome methylation data assessed in DNA extracted from saliva collected at baseline and 1 year. Cellular heterogeneity was accounted for in analyses. | Epigenetic age acceleration and GrimAge are positively associated with PTSD symptom severity at baseline and follow-up. Epigenetic age acceleration and GrimAge at follow-up are associated with cluster D symptoms. |
| Morris et al. (2022) | Longitudinal predictive study of young women in the USA with past 3-month experience of interpersonal violence | $n_{Total}$ = 58 $n_{PTSD}$ = 7 $nC_{ontrols}$ = 51 | With PTSD at 6 months: 25.0 ± 2.0 Without PTSD at 6 months: 23.7 ± 3.4 | 100% | White/ Caucasian 57%, Black/African American 33%, Asian 10%, Hispanic 7% | To construct biopsychosocial variable-based ML models to predict PTSD symptom trajectories and end-point PTSD status over a 6-month period | CAPS-IV at baseline, 1, 3 and 6 months | Diurnal and Trier Social Stress Test-elicited levels of cortisol and alpha amylase in saliva, and hair cortisol collected at baseline | Diurnal and Trier Social Stress Test-elicited cortisol and alpha amylase and hair cortisol were not associated with PTSD symptom trajectory in multilevel models but did improve predictive accuracy of ML models. |
| Nöthling et al. (2021) | Cross-sectional and longitudinal study examining epigenome-wide (discovery sample) and targeted (replication sample) DNA methylation in | Discovery $n_{Total}$ = 48 $n_{PTSD}$ = 24 Replication $n_{total}$ = 96 $n_{PTSD}$ = 39 | Discovery 25.9 ± 5.4 years Replication 24.6 ± 5.5 years | 100% | Black African 100% | To identify PTSD-associated methylation profiles in rape-exposed women; validate and replicate selected significant findings; and investigate the | DTS at baseline, 3 and 6 months | Epigenome-wide methylation using DNA extracted from blood samples at 3 months, and targeted methylation analyses using DNA extracted from | Cross-sectional analysis in discovery cohort identifies 1 DMP (cg01700569) and 34 DMRs associated with PTSD status at 3 months post-rape in the |

*Cambridge Prisms: Global Mental Health*

| Study | Design and setting | Sample characteristics | | | | Aims | Clinical measures | Molecular measures | Primary findings |
|---|---|---|---|---|---|---|---|---|---|
| | | Sample size | Age | Sex (female) | Ancestry/ethnicity | | | | |
| | rape-exposed South African females with and without PTSD | | | | | longitudinal association between methylation profiles of targeted genes and PTSD symptomology | | peripheral blood collected at baseline, 3 and 6 months. Cellular heterogeneity was accounted for in analyses. | discovery cohort. Targeted analysis of methylation in *BRSK2* and *ADCYAP1* performed but neither baseline nor longitudinal methylation profiles predict PTSD symptom severity when covariates are included in models. |
| Pereira et al. (2021) | Longitudinal study of offspring PTSD risk in 205 intimate partner violence-exposed mother-offspring dyads in the USA | 205 dyads of trauma-exposed mothers and their preschool children | Children: 4.64 ± 0.84 years | Mothers: 100% Children: 51% | African American 49%, Hispanic/Latinx 33%, European American 18%, other 1% | To examine whether childhood *FKBP5* genotype interacts with child trauma exposure to increase risk for PTSD and whether this genotype moderates the relationship between maternal and child PTSD symptoms | TSCYC (children) and PCL-C (mother) at baseline and 6–9 months | Child rs1360780 genotype measured in DNA extracted from saliva or buccal cells | Association between maternal and child PTSD symptoms was moderated by *FKBP5* genotype such that these symptoms were only associated among children with the minor T allele (CT/TT), but not the homozygous CC allele. |
| Schultebraucks et al. (2021) | Longitudinal predictive study of participants admitted to hospital emergency departments in Amsterdam (Netherlands) | $n_{Total} = 417$ $n_{Training} = 335$ $n_{Testing} = 82$ | 46.09 ± 15.88 years | 37.20% | Not specified but recruitment in the Netherlands with Dutch language proficiency required | To construct ML models for PTSD symptom trajectories over 12 months using biomedical data typically available upon hospital admission | IES-R at baseline, 1, 3, 6 and 12 months and CAPS-IV at 12 months | Physiological- and endocrine-related markers of sympathetic nervous system activity, hypothalamic–pituitary–adrenal axis, and hypothalamic–pituitary-thyroid axis activity, and information pertaining to DHEA levels | The 15 most influential variables in differentiating end-point PTSD status included thyroid stimulating hormone, cortisol, DHEA and free thyroxine. |
| Tamman et al. (2022) | Longitudinal study of US military veterans without PTSD recruited to the NHRVS study | n = 1,083 | 64.7 ± 12.5 years | 5.3% | European 100% | To evaluate how polygenic risk scores interact with social-environmental factors to predict incident PTSD | PCL-IV at baseline and PCL-5 at 2, 4 and 7 years | Polygenic risk score determined from whole-genome genotyping of DNA extracted from saliva samples | Higher polygenic risk score predicted increased risk of screening positive for PTSD over 7 years but only in participants with insecure attachment. |

*(Continued)*

| Study | Design and setting | Sample characteristics | | | | Aims | Clinical measures | Molecular measures | Primary findings |
|---|---|---|---|---|---|---|---|---|---|
| | | Sample size | Age | Sex (female) | Ancestry/ ethnicity | | | | |
| Vuong et al. (2022) | Longitudinal observational study of 455 rape-exposed South African women recruited to the RICE study | n = 455 | 25.3 ± 5.5 years | 100% | Black African 100% | To examine the association between *ADIPOQ* variants, alone and in interaction with childhood trauma, on PTSD symptom severity | DTS at baseline, 3 and 6 months | Genotyping of 8 *ADIPOQ* SNPs (rs16861194, rs16861205, rs17300539, rs2241766, rs6444174, rs822395, rs1402697 and rs1501299) using DNA extracted from whole blood | rs6444174TT genotype was associated with lower baseline PTSD symptoms in the unadjusted model. No genotype was associated with change in PTSD symptom severity over time and the genotype x childhood trauma interaction was not tested |
| Vuong et al. (2022) | Longitudinal observational study 1,135 rape-exposed and unexposed South African women recruited to the RICE study | $n_{Total}$ = 1,135 $n_{rape-exposed}$ = 542 $n_{unexposed}$ = 593 | Total: 25.2 ± 5.4 years Rape-exposed: 24.7 ± 5.3 years Unexposed: 25.7 ± 5.5 years | 100% | Black African 100% | To evaluate the relationship between adiponectin levels and probable PTSD risk, and to determine whether rape exposure moderates this relationship | MINI PTSD at baseline and DTS at 0, 3 and 6 months | Adiponectin assays performed using serum collected at baseline | Participants with mid to high adiponectin levels had a reduced risk of probable PTSD at 6 months follow-up that was independent of adiposity. There was no significant interaction with rape exposure. |
| Wani et al. (2021) | Prospective US population-based longitudinal cohort study of participants recruited to the DNHS study | n = 148 | 54.57 ± 12.79 | 60% | African American 93.8% | To apply ML models to psychopathology, social adversity, and glucocorticoid receptor regulatory network centred DNA methylation data to predict PTSD symptom severity | PCL-C collected across 5 waves | Genome-wide DNA methylation assessed using DNA extracted from 500 blood samples collected across different waves from 190 participants. Cellular heterogeneity was accounted for in analyses. | ML models identified that prior PTS symptom severity is the strongest predictor in ML models. 44 glucocorticoid receptor regulatory network DMPs were significantly associated with prospective PTSD symptom severity. |
| Wuchty et al. (2021) | Longitudinal predictive study of adults admitted to emergency units in two US hospitals | RNA group n = 366 Genotype and RNA group $n_{total}$ = 297 $n_{PTSD}$ = 108 | Genotype and RNA group 34.9 ± 12.7 years | 43% | Black/African American 69.9%, Caucasian 24.4%, Hispanic/ Latino 14% | To determine whether an integrative transcriptomic-genomic approach can highlight blood-based markers predictive of PTSD and/or major depressive disorder symptoms 6 months after admission to an emergency unit | PSS-I at 6 months | Transcriptome and eQTL analysis using blood collected at baseline. Control for sample cell composition not reported. | Study identified 13 driver causal genes and 458 paths from driver to corresponding dysregulated genes in PTSD. |

**Table 1.** (*Continued*)

| Study | Design and setting | Sample size | Age | Sex (female) | Ancestry/ ethnicity | Aims | Clinical measures | Molecular measures | Primary findings |
|---|---|---|---|---|---|---|---|---|---|
| | | **Sample characteristics** | | | | | | | |
| Yang et al. (2021) | Longitudinal study of US Combat veterans from the Operation Iraqi Freedom/Operation Enduring Freedom conflicts recruited to the PTSD Systems Biology Consortium study | Cross-sectional discovery $N_{Total}$ = 162 $n_{PTSD}$ = 80 Cross-sectional validation $n_{Total}$ = 53 $n_{PTSD}$ = 26 Longitudinal recall $N_{Total}$ = 55 | Cross-sectional discovery With PTSD: 32.7 ± 7.4 years Without PTSD: 32.5 ± 8.0 years Cross-sectional validation With PTSD: 36.9 ± 10.2 years Without PTSD: 34.0 ± 9.4 years Longitudinal recall With PTSD: 33.0 ± 7.5 years With subthreshold PTSD: 37 ± 8.2 years Without PTSD: 35.1 ± 7.5 years | 0% | Across cohorts: Hispanic 36%, non-Hispanic Asian 5%, non-Hispanic Black 23%, non-Hispanic White 32%, non-Hispanic other 7% | To investigate cross-sectional and longitudinal biological ageing in PTSD using an epigenetic clock estimate sensitive to age-related morbidity and mortality | SCID and CAPS-IV at baseline and three years | DNAm GrimAge epigenetic clock estimate using genome-wide DNA methylation profiles assessed in DNA extracted from blood at baseline and three years. Cellular heterogeneity was accounted for in analyses. | PTSD status was associated with advanced biological ageing, and CAPS score was positively associated with epigenetic age acceleration. Longitudinal analyses found a positive correlation between change in epigenetic age and change in PTSD symptom scores. |

*Note:* The systematic review is limited to original research manuscripts published from 1 January 2022 to 31 December 2023. The primary findings column only reports the results of analyses specifically assessing the relationships between PTSD symptomology and molecular measures. Participant age is reported as mean and standard deviation unless otherwise noted.

Abbreviations: 2-AG, 2-arachidonoylglycerol; Aβ40, amyloid β-40; Aβ42, amyloid β-42; AEA, N-arachidonoylethanalomine; STARRS, Study to Assess Risk and Resilience in Servicemembers; CAPS-5, Clinician-Administered PTSD Scale for DSM5; CAPS-IV, Clinician-Administered PTSD Scale for DSM-IV; CRP, C-reactive protein; DHEA, dehydroepiandrosterone; DMP, differentially methylated position; DMR, differentially methylated region; DNHS, Detroit Neighbourhood Health Study; eQTL, expression quantitative trait loci; GFAP, glial fibrillary acid protein; IES-R, Impact of Event Scale – Revised; IL-1β, interleukin-1 β; IL-10, interleukin-10; IL-6, interleukin-6; LASC, Los Angeles Symptoms Checklist; meQTL, methylation quantitative trait loci; MINI, Mini International Neuropsychiatric Interview; ML, machine learning; MRS, Marine Resilience Study; NFL, neurofilament light chain; NHRVS, National Health and Resilience in Veterans Study; NSE, neuron-specific enolase; PCL-5, PTSD Checklist for DSM5; PCL-C, PTSD Checklist civilian version; PCL-IV, PTSD Checklist for DSM-IV; pNfH, phosphorylated neurofilament heavy chain; PPQ, Perinatal PTSD Questionnaire; PRISMO, Prospective Research in Stress-Related Military Operations; PTS, posttraumatic stress; PTSD, posttraumatic stress disorder; PSS-I, PTSD Symptom Scale – Interview; RICE, Rape Impact Cohort Evaluation; SCID, Structured Clinical Interview for DSM; SRIP, Self-Rating Inventory for PTSD; TNFα, tumour necrosis factor α; TRACTS, Translational Research Center for TBI and Stress Disorders; TSCYC, Trauma Symptom Checklist for Young Children.

from T1 to T3, intrusive symptoms between T1 and T2, and avoidance symptoms between T2 and T3. Chen et al. (2022) found that the AA genotype of rs1800872, a single nucleotide polymorphism (SNP) in the gene encoding the anti-inflammatory cytokine interleukin-10 (IL-10), predicted PTSD symptom severity persistence between 6 and 18 months, as well as higher 18-month PTSD scores, in male but not female students following earthquake exposure.

In a study examining endocannabinoids, deRoon-Cassini et al. (2022) found that male and ancestry minority (predominantly African) participants admitted to hospital following traumatic injury who were homozygous for the A allele of the *FAAH* rs324420 SNP had higher baseline serum levels of N-arachidonoylethanalomine compared to C allele carriers. Elevated N-arachidonoylethanalomine positively predicted 6- to 8-month follow-up PTSD status across participants and avoidance and negative alterations of cognition and mood in female participants. Baseline circulating levels of 2-arachidonoylglycerol predicted follow-up reexperiencing, arousal and negative alterations to cognition and mood symptoms in ancestry minority participants. Endocannabinoid signalling is stress responsive and involved in a broad range of biological processes, such as neurotransmission, synaptic plasticity, inflammation, and learning and memory, which may be relevant to PTSD (deRoon-Cassini et al., 2022; Gorzkiewicz and Szemraj, 2018). Vuong et al. sought to identify whether the anti-inflammatory cytokine, adiponectin, may play a role in PTS symptom severity in women following rape exposure. Of the eight SNPs in *ADIPOQ,* the gene encoding adiponectin, that were investigated, only rs444174 was associated with symptom severity at 3 and 6 months though this relationship did not remain significant when models were adjusted for covariates (Vuong et al., 2022). In a study that included the same rape-exposed participants, as well as unexposed controls, lower baseline serum levels of adiponectin were associated with increased risk of probable PTSD at 6 months across participants. However, this relationship did not hold true when analyses were limited to women exposed to rape (Vuong et al., 2022).

*Polygenic risk score analyses*
Tamman et al. (2022) found that a higher polygenic risk score, an aggregate score reflecting genetic predisposition liability, predicted increased risk for an incident-positive PTSD screen at 2, 4 or 7 years in military veterans with an insecure attachment style. Of 30 genetic loci significantly associated with PTSD, 13 loci conferred risk or resilience based on patterns of environmental features (trauma burden, age, sex, social and structural support, and combat status). The strongest effect was seen for the rs4702 SNP in the gene encoding FURIN, which has been linked to brain-derived neurotrophic factor and matrix metalloproteinase signalling and plays a role in synaptic plasticity, including in relation to fear and environmental adversity. Risk score gene sets showed enrichment for immune function, specifically biological processes relevant to mast cells. Drug repositioning based on gene ontology highlighted doxylamine, an antihistamine and antimuscarinic that suppresses immune system activation and regulates sleep/wake cycles, as a therapy for investigation.

*Epigenetic investigations*
Several studies examined epigenetic mechanisms, that is, environmentally sensitive structural changes to DNA conformation that can influence gene expression (Aristizabal et al., 2019). Hawn et al. (2022) examined whether methylation in *AIM2*, a gene previously linked to inflammation, was associated with PTSD symptom severity and mediated the relationship between PTSD symptomology and markers of inflammation and neuropathology in a cohort of military veterans. The analyses indicated both time- and outcome-dependent effects of methylation. Lower methylation at the cg10636246 site mediated the baseline inverse association between PTSD symptom severity and the neuronal damage marker neurofilament light chain. However, lower cg10636246 site methylation mediated a positive association between symptom severity and proinflammatory (interleukin-6 and tumour necrosis factor alpha) as well as anti-inflammatory cytokines (IL-10) at baseline, as well as an inverse association between symptom severity and IL-10 2 years later. Drawing on measures of social adversity, prior psychopathology and methylation in sites linked to the glucocorticoid receptor response network, Wani et al. (2021) used machine learning (ML) to prospectively predict the risk of high PTSD symptom severity in an adult cohort. Though prior PTS symptoms accounted for 88% of the variance, CpG sites made up more than half of the top 150 model features and implicated stress response and inflammatory mechanisms in PTSD risk. Notably, methylation at cg20509117 in IL-6 was among the 79 sites identified. In a study investing intergenerational effects of intimate partner violence-associated PTSD, Cordero et al. (2022) aimed to assess correlations between maternal and infant site- and region-level glucocorticoid receptor (*NR3C1*) methylation profiles and to determine whether maternal methylation profiles predicted offspring internalising and externalising behaviours measured 4 years later. The average methylation of 13 glucocorticoid receptor promoter region sites was significantly correlated between mothers and their infants only in the context of PTSD, that is, not in control mother-infant dyads. A lower maternal methylation profile predicted increased externalising behaviour measured when children were of school-going age. These findings suggest that epigenetic signatures related to stress responding may contribute to the intergenerational effects of PTSD on mental health by biologically embedding adverse outcomes. In a cohort of military personnel with pre- and post-deployment molecular and clinical data, methylation at 4 sites and in 88 differentially methylated regions (DMRs) was associated with baseline PTS symptom severity, whilst longitudinal analysis identified 15 deployment-associated DMRs (Katrinli et al., 2022). The two DMRs predicting both baseline and longitudinal PTS symptom severity were located near genes encoding OTUD5 and ELF4, which are involved in inflammatory and oxidative stress processes. In a study investigating epigenetic signatures of PTSD following sexual assault, Nöthling et al. (2021) identified that baseline methylation at an intergenic site near the SLC16A9 gene was higher in rape-exposed women who met criteria for PTSD 3 months later. Two of the 34 PTSD-associated DMRs were located in genes (*BRSK2* and *ADCYAP1)* previously implicated in mood or trauma-related disorders. *BRSK2* is involved in neurotransmission and is highly expressed in hippocampus, a brain region fundamental to learning and memory, while the protein encoded by *ADCYAP1* plays a key role in stress response regulation. Targeted methylation analyses of regions in *BRSK2* and *ADCYAP1* were performed in a replication sample, and the longitudinal associations between PTSD symptom and methylation profiles at baseline, 3 and 6 months in both the discovery and replication samples were assessed. Of these follow-up analyses, only an association between changes from baseline to 3 months in PTSD symptom severity and

methylation at a *BRSK2* site remained significant after controlling for covariates.

Methylation data can be used to generate epigenetic clock estimates of biological ageing, that is, wear and tear that occurs over and above that due to chronological age (Horvath, 2013). Accelerated biological ageing according to DNAm GrimAge, a clock sensitive to age-related morbidity and mortality, in combat veterans was associated with PTSD status at baseline in both discovery and validation cohorts, as well as PTSD symptom severity at baseline when the cohorts were combined. In a subset of discovery cohort participants who completed assessments 3 years later, change in PTSD symptom severity was positively correlated with change in age acceleration, with this relationship primarily driven by symptoms in the hyperarousal cluster. The contribution of immune mechanisms to advanced age acceleration is uncertain given that this biological ageing metric was not associated with circulating levels of five cytokines but was significantly associated with increased immunosenescence based on CD8 and CD28 T lymphocyte proportions (Yang et al., 2021). A study of paramedicine students also found evidence for a relationship between PTSD symptomology and GrimAge estimates with both baseline and one-year GrimAge estimates positively associated with PTSD symptom severity at 1 year (Mehta et al., 2022). Baseline epigenetic age acceleration, a metric derived from the Horvarth algorithm, was positively associated with cross-sectional and longitudinal PTSD symptom severity, whilst baseline, but not follow-up symptom severity, predicted 1-year epigenetic age acceleration.

### Studies employing multimodal data

Four studies utilised multilevel data. Schultebraucks et al. (2021) applied ML models to data routinely collected on emergency room admission and in the 2 days thereafter (endocrine, trauma phenotype, demographic, vital signs, pharmacotherapy, and injury and trauma characteristics) to identify prognostic features for PTSD. The model identified that acute sympathetic nervous system activity, cortisol levels and opiate medication prescription, in addition to age, prior trauma experience and perceived impact of these events, perceived life threat, and amnesia predicted longitudinal (1-, 3-, 6- and 12-month) self-reported symptom severity and 12-month PTSD diagnosis. Three measures of thyroid function were among the 15 strongest predictors, suggesting a role for the hypothalamic–pituitary–thyroid axis. A particular strength of this study was its use of routinely collected data, which provides a higher potential clinical utility. Lori et al. (2021) found that baseline expression levels of *GRIN3B*, which encodes an NMDA glutamate receptor subunit, in blood were significantly associated with PTS scores at 1, 3, 6 and 12 months and showed a dose-dependent positive relationship with symptom trajectory (chronic vs. remitting vs. resilient) in individuals admitted to hospital emergency units. The authors examined whether methylation and expression quantitative trait loci, that is, loci where methylation or genetic variants, respectively, influence gene expression, could explain the observed relationship between *GRIN3B* expression and risk for chronic PTSD. The analysis yielded no significant methylation quantitative trait loci, but four SNPs were identified as *GRIN3B* expression quantitative trait loci with the minor allele of the rs10401454 SNP, which is associated with reduced expression, also associated with resilience to PTSD in an independent study cohort. *GRIN3B* encodes a subunit of the NMDA glutamate receptor, which may be upregulated in response to stress, and thus implicates glutamatergic processes, such as fear conditioning in the development of PTS symptoms. Furthermore, leukocyte expression of NMDA receptors suggests a tangential link to immune mechanisms in PTSD via *GRIN3B* effects on NMDAR activity. The study drew on the Genotype-Tissue Expression database to confirm that rs10401454 acts as an expression quantitative trait locus in whole blood, cerebellum and basal ganglia tissue and that minor allele homozygosity is associated with lower *GRIN3B* expression in cortex and frontal cortex. This indicates a degree of cross-tissue coherence and provides more confidence in inferring brain expression levels based on peripheral tissue measures. Wuchty et al. (2021) collected multiomic data from participants immediately and 6 months after trauma exposure. Based on the rationale that dynamic gene expression profiles provide valuable biological insights and that transcriptomic signatures could act as an intermediate phenotype, their study integrated genotype, transcriptomic and gene expression data. Using directed network analysis to draw causal inferences, they identified 13 genes that may drive dysregulated gene expression underlying PTSD development. Key processes supported by these genes are related to threat processing and responding, and fear-related memory acquisition, consolidation, and extinction. Morris et al. (2022) used multilevel modelling and ML to examine demographic, cognitive, clinical and biological (diurnal and laboratory stressor-elicited cortisol and alpha-amylase levels, as well as heart rate and hair cortisol) data to predict PTSD development at 1-, 3- and 6-month timepoints in young women exposed to interpersonal violence in the 3 months prior to study initiation. These factors were not associated with baseline or longitudinal PTS symptom severity using multilevel modelling but did improve ML model accuracy, suggesting that the stress response and acute sympathetic nervous system activity play an indirect role in the development of PTSD.

### Quality appraisal

All studies adequately described exposure and/or outcome measures, and most assessed potential distorting influences and clearly delineated control or comparison groups where appropriate (Table 2). Limitations were noted in the categories related to the description of the sample and reporting on data. Several studies had relatively modest sample sizes and most did not report on study power. Given the cost of conducting molecular-level investigations, especially those that draw on omics measures, this is not unexpected but does raise doubts as to whether the sample can be considered representative of the source population. The number of potential covariates or confounding influences assessed varied widely across studies. Imputation was frequently employed to address missing data, though several studies did not specifically address the presence/absence of missing data or how this was handled.

## Discussion

### Key findings

#### Molecular profiles

Several key themes emerge from the study findings. First, converging lines of evidence support the role of inflammatory/immune (Wani et al., 2021; Chen et al., 2022; Hawn et al., 2022; Katrinli et al., 2022; Tamman et al., 2022), stress response (Carleial et al.,

**Table 2.** Quality assessment of included studies according to the systematic appraisal of quality in observational research tool

| Study | Sample | Control/comparison group | Exposure/outcome measures | Distorting influences | Reporting of data | Overall study quality |
|---|---|---|---|---|---|---|
| Chen et al. (2022) | Highschool students exposed to the Wenchuan earthquake. Sampling method, representativity and appropriateness of sample size are unclear. (Quality: inadequate) | IL-10 rs1800872 genotypes. Control group is identifiable and source is clear. (Quality: adequate) | Exposure: social and environmental factors and IL-10 rs1800872 genotype. Outcome: PTSD symptom severity. Exposure and outcome measures adequately assessed. (Quality: adequate) | Demographic and environmental factors considered. (Quality: adequate) | No statement regarding missing data. Data clearly and accurately presented with results indicating whether test statistics reached different p-value thresholds. (Quality: unclear) | Moderate |
| Cordero et al. (2022) | Intimate partner violence-exposed women, with and without PTSD, and their offspring recruited to the Geneva Early Childhood Stress Study. Sample source, sampling method and inclusion/exclusion criteria are clear. Representativity and appropriateness of sample size are unclear. (Quality: unclear) | Women exposed to intimate partner violence without PTSD. Control group is identifiable and matched to cases, source is clear. (Quality: adequate) | Exposure: Maternal PTSD and *NR3C1* methylation. Outcome: Infant *NR3C1* methylation and child internalising and externalising behaviour. Exposure and outcome measures adequately assessed. (Quality: adequate) | Offspring demographic factors and maternal trauma- and stress-related variables considered. (Quality: adequate) | Missing data addressed in text. Data clearly and adequately represented with CIs and p-values. (Quality: adequate) | High |
| deRoon-Cassini et al. (2022) | Adults hospitalised for traumatic injury. Sample source and inclusion/exclusion criteria are clear and appropriateness of sample size is addressed. Sampling method and representativity are unclear. (Quality: unclear) | Minority status (not Caucasian, non-Hispanic), female sex, and SNP genotypes. Control group is identifiable and source is clear. Whether case–control differences controlled for is unclear. (Quality: unclear) | Exposure: genotype, 2-AG, AEA and plasma cortisol concentrations. Outcome: PTSD symptom severity. Exposure and outcome measures adequately assessed. (Quality: adequate) | Models stratified by sex and race/ethnicity. (Quality: Adequate) | Missing data addressed in text. Data clearly and adequately represented with p-values. (Quality: adequate) | Moderate |
| Hawn et al. (2022) | Cohort of US military veterans recruited to the TRACTS study. Inclusion/exclusion criteria are clear. Referenced paper provides detail on participant source and sampling method. Representativity and appropriateness of sample size are unclear. (Quality: unclear) | N/A | Exposure: Lifetime PTSD severity and baseline levels of inflammatory and neuropathology markers and *AIM2* methylation. Outcome measures: Follow-up inflammatory and neuropathology marker levels. Exposure and outcome measures adequately assessed. (Quality: adequate) | Demographic and cell type measures considered. (Quality: adequate) | Missing data addressed in text. Data clearly and adequately represented with p-values. (Quality: adequate) | High |
| Katrinli et al. (2022) | Military personnel/veterans recruited to three cohort studies: MRS (USA), Army STARRS (USA) and PRISMO (Netherlands). Sample source, sampling method and representativity are clear. Cohort-specific inclusion/exclusion | Trauma-exposed controls falling below study-specific PTSD severity score thresholds. Control group is identifiable, source is clear and case–control differences checked for. (Quality: adequate) | Exposure: Deployment and epigenome-wide DNA methylation. Outcome: PTSD symptom severity. Exposure and outcome measures adequately assessed. (Quality: adequate) | Demographic and methylation assessment-related variables considered. (Quality: adequate) | Missing data addressed in text. Data clearly and adequately represented with p-values. (Quality: adequate) | High |

*(Continued)*

**Table 2.** (*Continued*)

| Study | Sample | Control/comparison group | Exposure/outcome measures | Distorting influences | Reporting of data | Overall study quality |
|---|---|---|---|---|---|---|
| | criteria and adequateness of sample size unclear. (Quality: unclear) | | | | | |
| Landoni et al. (2022) | Pregnant women attending birth preparation courses. Source, sampling method and inclusion/exclusion criteria are clear. Representativity and adequateness of sample size are unclear. (Quality: unclear) | 5-HTTPLPR polymorphisms. Control group is identifiable and source is clear. (Quality: adequate) | Exposure: 5-HTTPLPR polymorphisms. Outcome: PTSD symptom severity. Exposure and outcome measures adequately assessed. (Quality: adequate) | Whether covariates or confounding variables were considered is unclear. (Quality: inadequate) | Missing data addressed in text. Data clearly and accurately represented with results indicating whether test statistics reached different p-value thresholds. (Quality: adequate) | Moderate |
| Lori et al. (2021) | Adult participants admitted to a hospital emergency unit. Source, sampling method and inclusion/exclusion criteria are clear. Representativity and appropriateness of sample size are unclear. (Quality: unclear) | N/A | Exposure: Transcriptome, eQTL and meQTL. Outcome: PTSD symptom trajectories. Exposure and outcome measures adequately assessed. (Quality: adequate) | Demographic and trauma- and methylation assessment-related variables considered. (Quality: adequate) | Missing data addressed in text but final sample size not explicitly stated for all individual analyses. Data are clearly and accurately represented with p-values provided. (Quality: unclear) | Moderate |
| Mehta et al. (2022) | Trauma-exposed paramedicine students. Source and sampling method are clear. Inclusion/ exclusion criteria, representativity and adequateness of sample size are unclear. (Quality: inadequate) | N/A | Exposure: psychological distress, PTSD symptom severity, professional quality of life, sense of organisational membership and social support. Outcome: Change in epigenetic age estimation. Exposure and outcome measures adequately assessed. (Quality: adequate) | Demographic, clinical and methylation assessment-related factors considered. (Quality: Adequate) | Missing data addressed in text. Data are clearly and accurately presented with p-values provided. (Quality: adequate) | Moderate |
| Morris et al. (2022) | Young women with experience of interpersonal violence. Source, sampling method and inclusion/exclusion criteria are clear. Representativity and appropriateness of sample size are unclear. (Quality: unclear) | N/A | Exposure: Biopsychosocial factors including sociodemographic, cognitive, clinical and trauma-related data. Outcome: PTSD status. Exposure and outcome measures adequately assessed. (Quality: adequate) | Range of factors are included in ML model and methods were employed to address model overfitting. (Quality: adequate) | Missing features imputed. Results are clearly and accurately represented with p-values provided. (Quality: adequate) | High |
| Nöthling et al. (2021) | Rape-exposed women recruited to the RICE study. Source and inclusion/exclusion criteria are clear and referenced paper provides detail on sampling method. Representativity and appropriateness of sample size are | Participants falling below study-specific PTSD severity score thresholds. Control group is identifiable and matched to cases, source is clear. (Quality: adequate) | Exposure: Epigenome-wide and targeted DNA methylation. Outcome: PTSD status and symptom severity. Exposure and outcome measures adequately assessed. (Quality: adequate) | Demographic and trauma- and methylation assessment-related variables considered. (Quality: adequate) | Missing data imputed. Results are clearly and accurately represented with p-values provided. (Quality: adequate) | High |

**Table 2.** (*Continued*)

| Study | Sample | Control/comparison group | Exposure/outcome measures | Distorting influences | Reporting of data | Overall study quality |
|-------|--------|--------------------------|---------------------------|-----------------------|-------------------|-----------------------|
| | unclear. (Quality: unclear) | | | | | |
| Pereira et al. (2021) | Trauma-exposed mothers recruited to the MAPS study and their children. Source, sampling approach and inclusion/ exclusion criteria are clear. Representativity and appropriateness of sample size are unclear. (Quality: unclear) | rs1360780 genotype. Control group is identifiable and source is clear. (Quality: adequate) | Exposure: child *FKBP5* genotype and maternal PTSD symptom severity. Outcomes: Child PTSD symptom severity. Exposure and outcome measures adequately assessed. (Quality: adequate) | Demographic and trauma-related variables considered. (Quality: adequate) | No statement regarding missing data. Data are clearly and accurately represented with p-values provided. (Quality: unclear) | Moderate |
| Schultebraucks et al. (2021) | Adults admitted to hospital emergency departments. Source, sampling method and inclusion/exclusion criteria are clear. Representativity and appropriateness of sample size are unclear. (Quality: unclear) | N/A | Exposure: 51 variables including demographic, endocrine, psychophysiological, pharmacotherapeutic, PTSD symptom severity and trauma- and injury-related data. Outcome: PTSD symptom trajectories status. Exposure and outcome measures adequately assessed. (Quality: adequate) | Range of factors were included in the ML model and appropriate methods were employed to refine parameter selection and address model overfitting. (Quality: adequate) | Missing features imputed. Results are clearly and accurately represented with p-values provided. (Quality: adequate) | Moderate |
| Tamman et al. (2022) | Military veterans without PTSD. Source, sampling procedure and representativity are clear. Referenced paper provides detail on inclusion/exclusion criteria. Adequateness of sample size unclear. (Quality = adequate) | Participants without an incident-positive screen for PTSD. Control group is identifiable, source is clear and case– control differences checked for. (Quality: adequate) | Exposures: polygenic risk score, attachment style and social network size. Outcome: PTSD status. Exposure and outcome measures adequately assessed. (Quality: adequate) | Demographic, clinical, and trauma-related variables considered. (Quality: adequate) | Missing data imputed. Results are clearly and accurately reported with CIs and p-values provided. (Quality: adequate) | High |
| Vuong et al. (2022) | Rape-exposed women recruited to the RICE study. Source and inclusion/exclusion criteria are clear and referenced paper provides detail on sampling method. Representativity and appropriateness of sample size are unclear. (Quality: unclear) | Genotype of eight *ADIPOQ* variants. Control group is identifiable and source is clear. (Quality: adequate) | Exposure: Eight *ADIPOQ* SNPs. Outcome: PTSD symptom severity. Exposure and outcome measures adequately assessed. (Quality: adequate) | Demographic and trauma-related variables considered. (Quality: adequate) | Missing data imputed. Results are clearly and accurately reported with p-values provided. (Quality: adequate) | High |
| Vuong et al. (2022) | Women with and without rape exposure recruited to the RICE study. Source and inclusion/ exclusion criteria are clear and referenced paper provides detail on sampling method. Representativity and appropriateness of sample size are | Low adiponectin tertile. Control group is identifiable, source is clear and case– control differences checked for. (Quality: adequate) | Exposure: Rape exposure and adiponectin levels. Outcome: Probable PTSD. Exposure and outcome measures adequately assessed. (Quality: adequate) | Demographic, clinical and trauma-related variables considered. (Quality: adequate) | Missing data imputed. Results are clearly and accurately reported with p-values provided. (Quality: adequate) | High |

**Table 2.** (*Continued*)

| Study | Sample | Control/comparison group | Exposure/outcome measures | Distorting influences | Reporting of data | Overall study quality |
|---|---|---|---|---|---|---|
| | unclear. (Quality: unclear) | | | | | |
| Wani et al. (2021) | Adult participants recruited to the DNHS. Adequateness of sample size is clear. Referenced paper provides detail on participant source and sampling method. Representativity and inclusion/exclusion criteria are not clear. (Quality: unclear) | N/A | Exposure: Psychological, social adversity and DNA methylation profiles. Outcome: PTSD symptom severity. Exposure and outcome measures adequately assessed. (Quality: adequate) | Cell type considered. Feature selection process adjusted to account for hidden confounders within ML models. (Quality: adequate) | Missing data imputed. Results are clearly and accurately represented. (Quality: adequate) | High |
| Wuchty et al. (2021) | Adults admitted to hospital emergency departments. Source, sampling method and inclusion/exclusion criteria are clear. Representativity and appropriateness of sample size are unclear. (Quality: uncertain) | Trauma-exposed controls. Control group is identifiable and source is clear. Whether case–control differences controlled for is unclear. (Quality: adequate) | Exposure: Transcriptomic and genomic profiles. Outcome: PTSD status. Exposure and outcome measures adequately assessed. (Quality: adequate) | Statistical approaches to identify and adjust for hidden sources of variation associated with traditional demographic variables used. (Quality: adequate) | Missing data addressed in text. Results clearly and accurately reported. (Quality: adequate) | High |
| Yang et al. (2021) | Military veterans recruited to cohorts included in the PTSD Systems Biology Consortium. Source and inclusion/exclusion criteria are clear and adequateness of sample size is addressed. Referenced paper provides detail on sampling method. Representativity is unclear. (Quality: adequate) | Military veterans without PTSD. Control group is identifiable, source is clear and case–control differences were checked for. (Quality: adequate) | Exposure: PTSD status and change in symptom severity. Outcome: GrimAge acceleration. Exposure and outcome measures adequately assessed. (Quality: adequate) | Demographic, clinical and immunological variables considered. (Quality: adequate) | No statement regarding missing data. Data is clearly and accurately reported with p-values provided. (Quality: unclear) | High |

Abbreviations: 2-AG, 2-arachidonoylglycerol; AEA, N-arachidonoylethanalomine; CIs, confidence intervals; DNHS, Detroit Neighbourhood Health Study; eQTL, expression quantitative trait loci; IL-10, interleukin-10; MAPS, Multidimensional Assessment of Preschoolers; meQTL, methylation quantitative trait loci; ML, machine learning; MRS, Marine Resilience Study; PRISMO, Prospective Research in Stress-Related Military Operations; PTSD, posttraumatic stress disorder; RICE, Rape Impact Cohort Evaluation; SNP, single nucleotide polymorphism; STARRS, Study to Assess Risk and Resilience in Servicemembers; TRACTS, Translational Research Center for TBI and Stress Disorders.

2021; Pereira et al., 2021; Schultebraucks et al., 2021; Wani et al., 2021; Cordero et al., 2022; Morris et al., 2022) and learning and memory processes (Carleial et al., 2021; Lori et al., 2021; Wuchty et al., 2021; Tamman et al., 2022) in PTSD pathophysiology with exploratory hypothesis-generating approaches utilising whole-genome transcription, epigenetic and genotype data showing enrichment for these mechanisms. Inflammatory processes may contribute to PTSD symptomology by affecting cognition, stress responding, and the structure and function of brain regions involved in affect, and learning and memory (Kim et al., 2020; Sumner et al., 2020). Two studies directly investigated inflammatory mechanisms. Though the functional effects of the IL-10 SNP rs1800872 investigated by Chen et al. (2022) is unknown, Hawn et al. (2022) found evidence that a more proinflammatory environment may contribute to PTSD with higher circulating levels of proinflammatory cytokines and lower levels of an anti-inflammatory cytokine predicting worse PTSD symptom severity at baseline and 2 years, respectively. Inflammation is also reciprocally related to the activity of the hypothalamus-pituitary–adrenal axis, which is activated upon threat perception, coordinates the stress response, and may affect cognition and mood (Leistner and Menke, 2020). The studies by Pereira et al. (2021) and Cordero et al. (2022) directly assessed components of the stress response system, namely *FKBP5* rs1360780 genotype and *NR3C1* methylation, respectively. Their findings suggest that the sensitivity and number of glucocorticoid receptors play a role in the pathophysiology of PTSD. This, in addition to findings that including cortisol levels improves predictive performance of ML models (Schultebraucks et al., 2021; Morris et al., 2022), is in keeping with a substantial body of evidence indicating that

dysregulation of the stress response is a key component of PTSD pathophysiology (Fischer et al., 2021). Inflammation and stress responding can accelerate biological ageing and thus underlie the findings of increased epigenetic clock age estimates in PTSD (Wolf and Morrison, 2017; Mehta et al., 2022). Identification of processes related to synaptic plasticity, and learning and memory may be explained by previous finding that individuals with PTSD fail to restrict fear responses to the environment, context or cues that elicited the trauma. Instead, overgeneralisation of fearful memories to non-specific targets, combined with reduced fear extinction, can produce the symptoms of hyperarousal and intrusion characteristic of PTSD (Ressler et al., 2022).

The studies included in this participant provide strong evidence for biological processes in PTS. However, the capacity for these biological correlates to predict relative risk or symptom trajectory is not clear. Though genotype investigations are beneficial in that they provide a static measure, PTSD is a polygenic disorder and the contribution of individual variants is likely small (Nievergelt et al., 2019). Therefore, the results of targeted studies, such as those by Chen et al. (2022), Pereira et al. (2021) and Landoni et al. (2022), which identified roles for specific variants in PTSD risk, may not have sufficient discriminative capacity for clinical use. Polygenic risk scores, as generated in Tamman et al. (2022), can explain more of the variance in outcome but do not necessarily perform well across populations of different ancestry. This limitation to their use in global contexts is exacerbated by the underrepresentation of diverse ancestries in psychiatric genetics research (Nievergelt et al., 2019). Other genomic approaches also have limitations. Studies utilising epigenetic and transcriptomic data report relative associations that is, the analyses are based on comparisons within study groups and do not provide set cut-off values that could be used to stratify individuals according to risk. Methylation and transcriptomic measures can differ according to tissue type and cellular composition, the latter of which most, but not all, of the studies reviewed stated that they controlled for. Studies also indicated that the influence of biological mechanisms may depend on environmental as well as demographic factors such as sex and ancestry (Chen et al., 2022; deRoon-Cassini et al., 2022; Tamman et al., 2022). A further limitation in identifying biological correlates with potential clinical utility is that PTSD can have diverse presentations with recent studies indicating the presence of multiple typologies such as threat reactivity, low-symptom, high-symptom and dysphoric classes (Campbell et al., 2020; Bucich et al., 2022). This phenotypic complexity may be mirrored in the biological processes conferring risk for specific symptom clusters, as was seen in the studies by Landoni et al. (2022), deRoon-Cassini et al. (2022) and Yang et al. (2021). Consequently, it is also possible that existing and valid relationships between biological correlates and symptom typologies were not discerned in studies that used only PTSD status and/or total symptom scores. Based on these limitations, biological correlate research at present seems better able to inform on underlying mechanisms than to stratify individuals according to risk and target preventative measures and interventions.

### Study contexts

Selected studies were mainly conducted in UICs and comprised predominantly Caucasian ancestry participants, although those recruiting patients admitted to hospital emergency units and military veterans/personnel reflected a more diverse ancestry profile. Despite being home to 84.5% of the global population in 2021 (World Bank, n.d.), less than one-quarter of the selected studies were conducted in LMICs. These investigations reflect a more constrained participant demographic profile (Black adult females or Chinese male and female adolescents), range of trauma experiences (rape or earthquake exposure) and number of molecular techniques employed (candidate gene, inflammatory marker or DNA methylation) than UIC studies. None of the studies generating multimodal data or using ML or multilevel modelling were performed in LMICs. Representation within LMIC studies was also unequal with three of four publications based on a single South African cohort study examining outcomes of rape exposure in female participants that identified as Black African. Our search criteria did not yield any studies conducted in South American or South or South-East Asian populations. Furthermore, all but one study recruited adult participants. This is problematic given that the age structure in LMICs is skewed towards younger age categories and that trauma exposure occurring during sensitive windows of neurodevelopment can have substantial effects on neurobiology and mental health risk across the lifespan (Herringa, 2017). The treatment gap is higher in LMICs with treatment-seeking rates in these countries only half that reported in UICs, and established risk factors for PTSD, including social disadvantage, younger age, lower household income, unemployment and lower levels of education, are observed at disproportionately higher rates in LMICs (Koenen et al., 2017). These LMIC vs. UIC disparities undermine efforts to address the global burden of PTSD. Understanding the interaction of environmental and molecular neurobiological determinants in the genesis of PTSD is key to the discovery of new and effective treatments, which has been highlighted as a particularly pressing need in LMICs. However, the potential for biological mechanisms to inform preventative and therapeutic measures will only be realised if the number and scope of studies more closely align with the global burden of trauma and PTSD.

### Study approaches

Several studies integrated multimodal data which can provide comprehensive and deep insights into psychiatric outcomes (Wuchty et al., 2021). Studies made use of sophisticated analyses, including ML and multilevel modelling. The former does not test specific hypotheses but rather learns which combination of features best predicts outcomes and, by including variable interrelationships, can account for indirect contributions to outcomes. In contrast, multilevel modelling assesses how variables are related and can thus provide a more interpretable, but potentially less accurate, model (Morris et al., 2022). Unfortunately, the cost of collecting multilevel data limits study replication and sample size, the latter of which reduces the likelihood of adequately representing the source population or conducting sex-, age- or ancestry-stratified analyses. Therefore, it is important that LMICs are included in psychiatric genetic and other consortia of multi-country investigators and studies. Despite relatively small sample sizes, omics studies can increasingly draw on available public and consortia databases to interpret and gain functional insights into data. Examples include functional enrichment for biological processes and information on tissue-specific gene expression. The latter is particularly useful in neuroscience, where the inaccessibility of brain tissue necessitates inferences about neural mechanisms based on peripheral tissues. Though the results obtained from multiomics studies are informative, their clinical utility is low insofar as they rely on data that is not routinely collected from or available for individuals seeking post-trauma care (Schultebraucks et al., 2021).

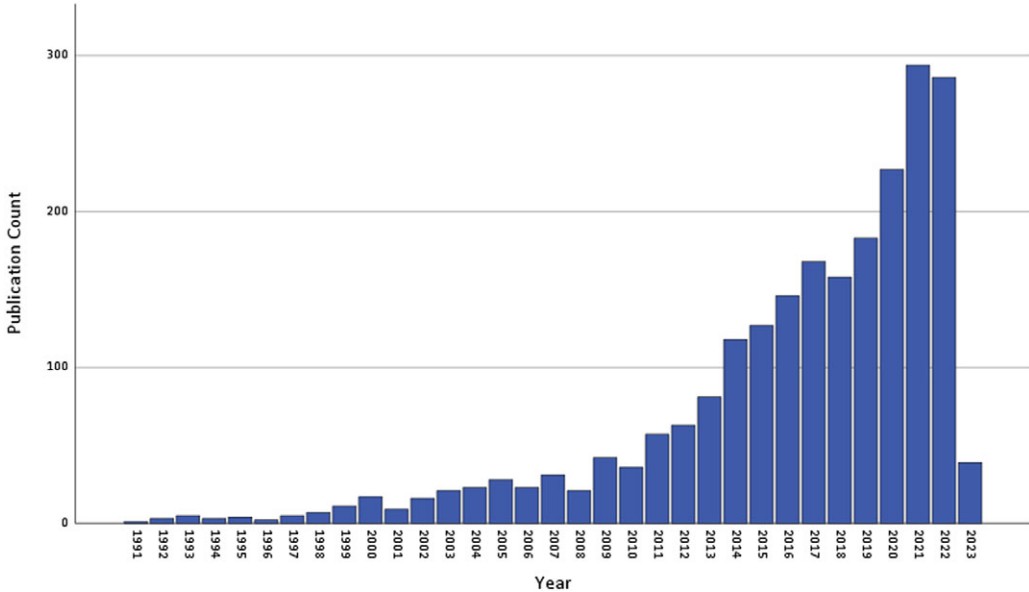

**Figure 2.** Annual count of number of publications retrieved from PubMed using the systematic review search string. The results are based on the following query: *(("post-traumatic stress" (All Fields)) OR ("posttraumatic stress" (All Fields)) OR (PTSD (All Fields))) AND ((neurobiolog* (All Fields)) OR (genom* (All Fields)) OR (DNA (All Fields)) OR ("stress hormone" (All Fields))) AND (English(Language)) NOT (animal).* The search was limited to studies published up to 31 December 2022.

## Limitations

This review has several limitations. We selected a narrow publication timeframe and limited our selection to longitudinal studies reporting results in English. This excludes insights offered by studies published in other languages and by cross-sectional investigations, and may have limited the representation of selected studies. Though the number of publications fitting our search terms is increasing year-on-year (Figure 2), it is not possible to discern whether this is true for both UICs and LMICs, or to draw inferences about trends in methodological approaches. The narrow timeframe, as well as the impact of COVID-19, could also have exaggerated the fault line between published research in UICs and LMICs. Our focus on molecular-level aetiology only reflects a subset of biological mechanisms and ignores findings from neuroimaging and psychophysiological studies. We did not report on animal model investigations, which have the potential to unravel causal mechanisms, as opposed to the correlative associations identified in human studies. We also did not conduct a meta-analysis of study findings. Finally, we did not consider the full range of trauma-associated adverse outcomes or the contribution of molecular profiles to cross-disorder risk.

## Conclusions

The potential benefits of biological insights into PTSD are compelling with recent research identifying inflammatory, stress response, and learning and memory processes in PTSD pathophysiology. The sophistication and scope of molecular techniques and data analytic approaches are rapidly expanding. However, greater representation of LMICs in research is required for biological insights to reduce the global burden of PTSD and associated adverse effects on health and wellbeing.

**Open peer review.** To view the open peer review materials for this article, please visit http://doi.org/10.1017/gmh.2023.53.

**Data availability statement.** A systematic review of published literature was performed. No original research data were generated.

**Author contribution.** J.S.W.: Conceptualisation, investigation, data curation, writing – original draft, writing – review and editing. M.P.: Investigation, data curation, writing – original draft, writing – review and editing. M.C.G.: Investigation, data curation, writing – original draft, writing – review and editing. L.L.H.: Conceptualisation, writing – review and editing. E.K.: Writing – review and editing. S.S.: Conceptualisation, writing – review and editing.

**Financial support.** Research reported in this publication was partly supported by the South African Medical Research Council. The work by JSW was made possible through funding by the South African Medical Research Council through its Division of Research Capacity Development under the Early Investigators Programme from funding received from the South African National Treasury. The content hereof is the sole responsibility of the authors and does not necessarily represent the official view of the SAMRC. MCG was supported by a Career Development Award from the U.S. National Institute of Mental Health (K01MH129572).

**Competing interest.** The authors declare no competing interests exist.

**Ethics standard.** All authors declare that they have adhered to the publishing ethics of Global Mental Health.

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
