## [Reviewer Report]

Dear editor,

On behalf of my co-authors, I am pleased to submit the following invited review "Advances in the molecular neurobiology of posttraumatic stress disorder from global contexts: A systematic review of longitudinal studies” to Global Mental Health.

Trauma exposure is a globally prevalent phenomenon that can exert profound and lasting effects on health and wellbeing. Nevertheless, only a subset of exposed individuals will go on to develop posttraumatic stress disorder, which is characterised by intrusive thoughts, avoidance behaviours, hypervigilance and negative alterations in cognition and mood. A substantial body of research indicates that biological mechanisms play an important role in determining risk and resilience, and advances in molecular technologies and data analytic procedures now provide unprecedented insights into the molecular aetiology underlying PTSD. These findings have the potential to reduce the burden of posttraumatic stress disorder on global mental health. Specifically, identification of biological correlates of posttraumatic stress disorder could be used to stratify trauma-exposed individuals according to risk, target prevention and intervention initiatives to those at highest risk, identify biological targets for potential therapeutic intervention and track treatment response.

In this systematic review, we aimed to summarise the findings of studies investigating the longitudinal relationships between molecular-level measures and posttraumatic stress disorder status or symptom severity. We chose to focus on studies published in 2021 and 2022 so as to provide a state-of-the-art overview. The 28 studies that met our inclusion criteria recruited participants with diverse experiences of trauma, including military conflict, interpersonal violence, sexual assault, traumatic injury and natural disasters. We provide an overview of the study results and find strong evidence for inflammation/immune, stress response and learning and memory processes as key biological signatures in the pathophysiology of posttraumatic stress disorder. We have highlighted the methodological advances in the field. Our manuscript also contains an embedded discussion of the nature and form of the literature in upper- compared to lower- and middle-income countries. The majority of studies included in the systematic review, and especially those that used multilevel data, were conducted in upper-income countries, which indicates a mismatch between trauma research and the global burden of trauma exposure and associated outcomes.

We believe that our manuscript is well suited to the aims and scope of the journal, as it provides an accessible yet comprehensive overview of recent research into the molecular neurobiology of posttraumatic stress disorder, a globally prevalent psychiatric disorder. We would like to thank you for granting us permission to exceed the 4000 word limit, which has allowed us to provide a more detailed overview of a substantial body of research. We would also like to confirm that this manuscript has not been previously published and is not under consideration by another journal.

We hope that you consider our manuscript favourably.

Sincerely,

Jacqueline Womersley

---

## [Reviewer Report]

Womersley and colleagues conducted a systematic review on longitudinal studies regarding the molecular neurobiology of PTSD. In their review, they report on findings from longitudinal studies investigating a wide range of biological parameters (e.g., genetic, epigenetic, and multimodal data) as well as samples, trauma types and types of study (e.g. PTSD trajectories and treatment studies).

They report the important finding, that, while PTSD rates are higher in LMICs compared to UICs, these countries are underrepresented in neurobiological research on PTSD.

My major points are the following:

1) The authors mention and discuss so many diverse studies that there is only little time to report each study and to put the results in a broader context. For instance, the authors mention at the beginning that longitudinal neurobiological research might guide patient stratification, however, for the reader it is difficult to tell how the reviewed studies contribute to this endeavor.

2) In this line, I would strongly recommend to focus on either predicting PTSD symptom trajectories OR to focus on treatment response, to increase the focus of the manuscript and to make it easier to draw meaningful conclusions from the presented data.

3) As already stated by the authors, the time frame of the included studies is relatively narrow. This is especially important as the authors report on the evidence gap from LMICs. Given the limited time frame, it is difficult to estimate if the numbers of studies truly reflect the extent of the problem. Further, it remains unclear if the numbers of studies from LMICs are decreasing or increasing or remain stable. Further, the COVID pandemic might have had an impact on research output from LMICs and UICs (and the reported period is likely still affected by the extent of research activities during the pandemic).

4) The introduction does not capture the global mental health focus of the article. For instance, in the introduction, data regarding trauma and PTSD prevalence from the US are reported, while data on much higher prevalence rates from LMICs is not mentioned here. I would suggest to include more evidence regarding the burden of conflict-related mental health problems in LMICs in the introduction (e.g. Charlson et al., 2019, the Lancet).

---

## [Reviewer Report]

The manuscript is well written and gives a nice overview of the current state of research. Relevant literature has been cited.

Minor comments:

1. since DNA methylation as well as gene expression are tissue specific, it should be stated whether and how cell composition was determined in the studies.

For example, for whole blood, was there a determination of cell number and cell type in the literature cited? Was a correction for the heterogeneous tissue performed? This information could be added to Table 1.

.

2. If this information is missing in one or more studies, this limitation should be mentioned in the discussion.

---

## [Reviewer Report]

Dear Drs Bass and Romero,

Thank you for considering our manuscript, “Advances in the molecular neurobiology of posttraumatic stress disorder from global contexts: A systematic review of longitudinal studies”.

We have carefully revised the manuscript in line with the reviewer comments and believe that this has improved the value of the paper. The major revisions to the manuscript are that the included articles have been limited to those examining biological measures as predictors of PTSD status or symptom severity, which has allowed us to provide more detail for each of the included studies, and we have also provided more information on the utility of biological correlates.

Thank you for inviting us to submit this review, and also for allowing us some leeway with the word limits. We hope that you will consider the revised manuscript favourably and look forward to hearing the outcome of our submission.

Sincerely,

Jacqueline Womersley

---

## [Reviewer Report]

The authors carefully revised the manuscript according to the feedback of the reviewers. I would recommend to accept the paper in ints current version.

---

## [Reviewer Report]

All comments were addressed and the manuscript revised accordingly. Three studies were removed from the analysis with explanation.